# SOLVING CONTINUAL OFFLINE REINFORCEMENT LEARNING WITH DECISION TRANSFORMER

## ABSTRACT

Continuous offline reinforcement learning (CORL) combines continuous and offline reinforcement learning, enabling agents to learn multiple tasks from static datasets without forgetting prior tasks. However, CORL faces challenges in balancing stability and plasticity. Existing methods, employing Actor-Critic structures and experience replay (ER), suffer from distribution shifts, low efficiency, and weak knowledge-sharing. To address these issues, we first compare AC-based offline algorithms with Decision Transformer (DT) within the CORL framework. DT offers advantages in learning efficiency, distribution shift mitigation, and zero-shot generalization but exacerbates the forgetting problem during supervised parameter updates. We introduce multi-head DT (MH-DT) and low-rank adaptation DT (LoRA-DT) to mitigate DT's forgetting problem. MH-DT stores task-specific knowledge using multiple heads, facilitating knowledge sharing with common components. It employs distillation and selective rehearsal to enhance current task learning when a replay buffer is available. In buffer-unavailable scenarios, LoRA-DT merges less influential weights and fine-tunes DT's decisive MLP layer to adapt to the current task. Extensive experiments on MoJuCo and Meta-World benchmarks demonstrate that our methods outperform SOTA CORL baselines and showcase enhanced learning capabilities and superior memory efficiency.

## 1 INTRODUCTION

Continuous offline reinforcement learning (CORL) (Gai et al., 2023) is an innovative paradigm that merges the principles of continuous learning with offline reinforcement learning. CORL aims to empower agents to learn multiple tasks from static offline datasets and swiftly adapt to new, unknown tasks. A central and persistent challenge in CORL is the delicate balance between plasticity and stability (Khetarpal et al., 2022). On one hand, the reinforcement learning policy must preserve knowledge and prevent forgetting of historical tasks (stability). On the other hand, it should exhibit the ability to rapidly adapt to new tasks (plasticity).

Existing methods predominantly integrate offline algorithms based on actor-critic structures with continual learning techniques (Gai et al., 2023), with rehearsal-based methods (Lopez-Paz & Ranzato, 2017; Chaudhry et al., 2018) being the most commonly employed. However, these methods encounter several challenges, including multiple distribution shifts, suboptimal learning efficiency, and limited knowledge-sharing capabilities. These distribution shifts manifest in three forms. Firstly, there is a distribution shift between the behavior policy and the learning policy inherent to AC-based offline algorithms. Secondly, distribution shifts occur between the offline data from different tasks, leading to catastrophic forgetting. Lastly, distribution shifts between the learned policy and the saved replay buffer which result in performance degradation in previous tasks during the rehearsal process. Regarding knowledge-sharing capabilities, while the relevance of the Q function in related tasks has been established (Niekerk et al., 2019), these methods merely introduce behavioral clones to the actor which shares few knowledge since the actor's objective is to maximize the Q value.

To solve the above problems, we first rethink the process of CORL by comparing an AC-based offline RL algorithm with Decision Transformer (DT) (Chen et al., 2021), another offline RL paradigm. The results underscore several advantages of DT, including heightened learning efficiency, bypassing distribution shift of offline learning, and superior zero-shot generalization capabilities. However, the forgetting issue of DT is more serious, manifested by a rapid decline in perfor-

mance after switching tasks. This heightened sensitivity to distribution shifts between time-evolving datasets is attributed to DT's training using supervised learning and updating all parameters.

In order to retain the advantages of DT and solve the more serious problem of catastrophic forgetting, we propose multi-head DT (MH-DT) and low-rank adaptation DT (LoRA-DT). MH-DT uses multiple heads to store task-specific knowledge and share knowledge with common parts, to avoid the catastrophic forgetting problem caused by all parameters being changed when the dataset distribution shift occurs. Besides, using the structural characteristics of the transformer and the feature that DT will benefit from training on close tasks we propose an additional distillation objective and selective rehearsal module to improve the learning effect of the current task. To solve the CORL problem when the replay buffer is not allowed – for example, in real-world scenarios where data privacy matters (Shokri & Shmatikov, 2015), inspired by Lawson & Qureshi (2023) that explored the similarities and differences of each module in the DT structure under multi-task situations, we proposed LoRA-DT that merges weights that have little impact for knowledge sharing and fine-tunes the decisive MLP layer in DT blocks with LoRA to adapt to the current task. This method also avoids substantial performance deterioration with a smaller buffer size (Cha et al., 2021).

Extensive experiments on MuJoCo (Todorov et al., 2012) and Meta-World (Yu et al., 2020a) benchmarks demonstrate that our methods outperform SOTA baselines in all CORL metrics. Our DT-based methods also show other advantages including stronger learning ability and more memory-efficient. The main contributions of this paper can be summarized as four folds:

- We propose that compared to the offline RL methods employing the Actor-Critic structure, methods utilizing DT as the foundational model are better suited for addressing the CORL problem and point out the advantages of DT and problems that need to be solved by rethinking the CORL process with decision transformer. To the best of our knowledge, we are the first to propose using DT as an underlying infrastructure in CORL setting.

- When the replay buffer is available, we propose MH-DT to solve the problem of catastrophic forgetting and avoid the problem of low learning efficiency by distillation and selective rehearsal.

- We proposed LoRA-DT that uses weights merging and low-rank adaptation to avoid catastrophic forgetting and save memory by saving the fine-tuned updated matrix when buffer is unavailable.

- We experimentally demonstrate that our methods outperform prior CORL methods, perform better learning capability and are more memory-efficient.

## 2 RELATED WORK

**Offline reinforcement learning.** Offline reinforcement learning allows policy learning from data collected by arbitrary policies, increasing the sample efficiency of RL. The key issues in offline reinforcement learning are distributional shift and value overestimation. Some prior works propose to constrain the learned policy towards the behaviour policy by adding KL-divergence (Peng et al., 2021; Nair et al.; Wang et al., 2020), MSE (Dadashi et al., 2021), or the regularization of the action selection (Kumar et al., 2019). Other works (Yu et al., 2020b; 2021) propose to train a dynamic model to predict the values of OOD samples in a supervised learning way. However, all these methods are based on AC structure, and their final performance depends on the accuracy of Q-value estimation. These methods cannot achieve good results in offline continuous reinforcement learning since it's difficult to obtain accurate estimates when distribution shifts and there is no obvious relationship between the actor-network of different tasks. Recently, Chen et al. (2021) proposed Decision Transformer(DT) to solve offline reinforcement learning problems by casting RL problem as conditional sequence modelling. DT demonstrates superior learning efficiency compared to AC-structured algorithms. In this paper, we propose that DT is more suitable for offline continuous reinforcement learning scenarios, use DT as a backbone network and focus on solving its catastrophic forgetting problem.

**Continual Reinforcement Learning.** Continual learning is a challenging and important problem in machine learning, where the goal is to enable a model to learn from a stream of tasks without forgetting the previous ones. Generally, continual learning methods can be classified into three categories (Parisi et al., 2019): regularization-based approaches (Kirkpatrick et al., 2017; Zenke et al., 2017) add a regularization term to prevent the parameters from far from the value learned from past tasks; modular approaches (Fernando et al., 2017; Mallya & Lazebnik, 2018) consider

fixed partial parameters for a dedicated task; and rehearsal-based methods (Chaudhry et al., 2018; Lopez-Paz & Ranzato, 2017) train an agent by merging the data of previously learned tasks with that of the current task. These methods mainly considered online reinforcement learning. In CORL problem setting, Gai et al. (2023) demonstrated that Experience Replay (ER), a rehearsal-based method, is the most effective. However, ER methods need to consider the stability-plasticity trade-off. The previous ER method was mainly based on AC structure and focused more on stability. As a result, there are too many tasks that need to be reviewed later in the learning process, resulting in failure to achieve good results in subsequent tasks. In this paper, we propose MH-DT, a method that applies a modular approach and experience replay method to DT, plus distillation objective and selective rehearsal for better plasticity. Then, considering the characteristics of DT in handling multi-tasks, we propose a new DT-based continuous learning method referred to as LoRA-DT, that uses weight merging to share knowledge and saves LoRA matrices to avoid forgetting.

## 3 RETHINKING CORL WITH DECISION TRANSFORMER

Below, we first review CORL problem setting and Decision Transformer. Then, we rethink the process of CORL using DT and compare it with offline algorithms using AC structure.

### 3.1 PRELIMINARY

**Continual Offline Reinforcement Learning** In this paper, we investigate CORL, which learns a sequence of RL tasks $\mathbb{T} = \{T_1, \cdots, T_N\}$. Each task $T_n$ is described as a Markov Decision Process (MDP) represented by a tuple of $\{\mathcal{S}, \mathcal{A}, P_n, \rho_{0,n}, r_n, \gamma\}$, where $\mathcal{S}$ is the state space, $\mathcal{A}$ is the action space, $P_n : \mathcal{S} \times \mathcal{A} \times \mathcal{S} \leftarrow [0,1]$ is the transition probability, $\rho_{0,n} : \mathcal{S}$ is the distribution of the initial state, $r_n : \mathcal{S} \times \mathcal{A} \leftarrow [-R_{\max}, R_{\max}]$ is the reward function, and $\gamma \in (0,1]$ is the discounting factor. We assume that sequential tasks have different $P_n, \rho_{0,n}$ and $r_n$, but share the same $\mathcal{S}, \mathcal{A}$, and $\gamma$ for simplicity. The return is defined as the sum of discounted future reward $R_{t,n} = \sum_{i=t}^{H} \gamma^{(i-t)} r_n(s_i, a_i)$, where $H$ is the horizon. In an online RL setting, the experiences $e = (s, a, s', r)$ can be obtained through environment interaction. However, in offline RL setting, the policy $\pi_n(a \mid s)$ can only be learned from a static dataset $\mathcal{D}_n = \{e_n^i\}, e_n^i = (s_n^i, a_n^i, s_n'^i, r_n^i)$, which is assumed to be collected by an unknown behavior policy $\pi_n^\beta(a \mid s)$.

**Decision Transformer (DT)** for offline RL treats learning a policy as a sequential modeling problem. It proposes to model trajectories with state $s_t$, action $a_t$ and reward-to-go $\hat{r}_t$ tuples collected at different time steps $t$. The reward-togo is the cumulative rewards from the current time step till the end of the episode. Instead of including the one-step reward $r_t$, this novel representation helps guide action selection towards optimizing the return. At timestep $t$, Decision Transformer takes a trajectory sequence $\tau$ autoregressively as input which contains the most recent $K$-step history $\tau = (\hat{r}_{t-K+1}, s_{t-K+1}, a_{t-K+1}, \ldots, \hat{r}_t, s_t, a_t)$. When training with offline collected data, $\hat{r}_t = \sum_{i=t}^{T} r_i$. During testing, $\hat{r}_t = G^\star - \sum_{i=0}^{t} r_i$ where $G^\star$ is the targeted total return for an episode. Each trajectory $\tau$ corresponds to $3K$ tokens in the standard Transformer model. To encode the sequence timestep information, DT concatenates the same timestep embedding to the embeddings of $s_t, a_t$ and $\hat{r}_t$. DT is trained to predict an action by minimizing mean-squared loss.

### 3.2 RETHINKING CORL WITH DECISION TRANSFORMER

We conduct experiments in the Cheetal-Vel environment, training Vanilla DT alongside an offline algorithm TD3+BC (Fujimoto & Gu, 2021) based on the AC structure. We use six tasks and train 30K steps for each task as in Fig.1. We hope to explore the following two questions:

**In CORL setting, what are the advantages of DT compared with AC structure algorithm?**

- DT's learning efficiency is higher than that of AC mode algorithms, and it can learn better-performing policies with the same data quality. This is even more necessary in a continuous learning setting, because in limited training steps, in addition to learning new tasks, operations such as distillation also need to be performed to achieve knowledge transfer. Faster single-task learning efficiency is more conducive to subsequent transfer.

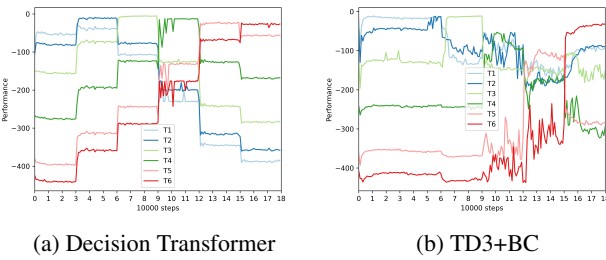

(a) Decision Transformer      (b) TD3+BC

Figure 1: Performance on each task during continuous learning.

- DT exhibits a distinctive capability in task identification. Notably, when there is a shift in the training dataset, DT demonstrates an immediate adjustment in its performance, corresponding to the change in tasks. This observation underscores its inherent ability for implicit task identification. In contrast, offline algorithms tend to exhibit more erratic learning curves.

- The model's performance can benefit from similar tasks (e.g. Performance on vel-26 keeps improving since the current task is getting closer), which gives us the possibility to review selectively.

- DT has a powerful memory ability and generalization ability in disguise, which is exactly what we need in the CORL setting. In steps 90K to 120K in Fig.1, DT has better zero-shot generalization performance in several untrained tasks.

**What problems do we need to solve when applying DT to CORL?** DT can identify changes in offline data distribution faster and quickly converge to a better policy under the new task, which also means that the forgetting problem of DT is more serious. We attribute this to the fact that DT is trained through supervised learning, and the vanilla DT updates all network parameters during training. Therefore, DT with history trajectory $\tau$ as input can more accurately perceive changes in the distribution of offline datasets, which are then reflected in changes in action output. In contrast, AC-structured methods typically follow a two-step process. They initially correct the estimation of Q value using temporal difference learning and subsequently adjust the parameters of the actor network to maximize Q. This process renders AC-based methods less susceptible to changes in the offline data distribution compared to DT. Simultaneously, DT's superior zero-shot generalization capability indicates its capacity for shared knowledge between tasks. Our objective, therefore, is to preserve and harness this shared knowledge across tasks while concurrently updating and preserving task-specific knowledge. This approach facilitates adaptation and guards against forgetting.

## 4 METHODOLOGY

Inspired by the above observation and analysis in Sec.3.2, we further propose MH-DT and LoRA-DT, two new DT-based methods that leverage the strengths and solve DT's more serious forgetting problem within CORL setting when the replay buffer is available or not respectively.

### 4.1 MH-DT: REPLAY BUFFER BASED CORL

Prior rehearsal-based approaches utilize a multi-head policy network $\pi$ and a Q-network $Q_n$ to learn task $T_n$, which means, during learning, policy network $\pi$ has two objectives: on the one hand, the multi-head policy $\pi$ is optimized for all current and previous tasks $T_1$ to $T_n$, for action prediction; on the other hand, the policy $\pi$ is also used for updating the current Q-network $Q_n$. This dual role of the policy network can lead to a significant performance decline in the rehearsal phase. While Gai et al. (2023) solves the inconsistency between the learning and the review objectives through the introduction of an intermediate policy $\mu_n$, it also presents a new challenge. Directly cloning multiple action distributions into a multi-head policy is meaningless and unexplainable because there is a correlation between the Q functions of different tasks, but no obvious relationship between the policy networks, which are designed to maximize Q. Experiments demonstrate that although it effectively learns a high-performing $\mu_n$, $\pi$ struggles to learn a policy that performs well on $T_n$ as

review tasks increase. Additionally, the Rehearsal process can lead to performance degradation due to a distribution shift between the saved trajectory and the learned policy.

Based on this analysis, we introduce Multi-Head Decision Transformer (MH-DT). This approach leverages DT to circumvent the Q-function learning step, mitigating issues associated with inaccurate Q-value estimation in offline settings. By employing supervised learning directly for training, we eliminate the need to address the distribution shift between the learned policy and saved data. Consequently, we can readily designate a portion of the prior offline dataset, $D_n$, as a replay buffer, denoted as $B_n$. The schematic diagram of our proposed architecture is given in Fig.2. The intermediate transformer module is used to learn shared environment knowledge, whose parameters are denoted as $\theta_z$. Each task possesses its dedicated head $h_n$ to store task-specific information whose parameters are denoted as $\theta_n$. Each head $h_n$ comprises two components: embedding layers and a layer-norm layer before the common module, plus a linear network responsible for action prediction following the common module. We denote $\pi_n$ as the network with combined parameters $[\theta_z, \theta_n]$ specific to task $T_n$ for evaluation. $\pi$ represents the entire MH-DT.

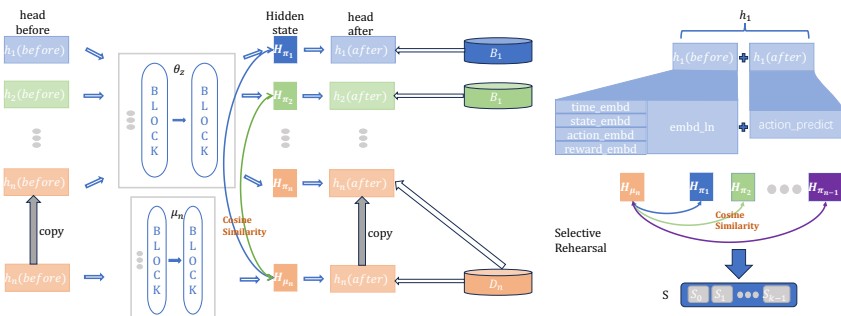

Figure 2: Schematic diagram of MH-DT. The left part is the training process. We first learn a separate policy $\mu_n$, copy the parameters of the head part to head $h_n$, then calculate the loss in Eq.(6) through the data in replay buffer $B_1, \ldots, B_{n-1}$ and $D_n$, and update the corresponding head and shared parameters. The upper right part is the structure of each head. The front part includes embedding layers and a layer-norm layer, and the back part includes a linear layer for predicting actions. The lower right part is the schematic diagram of task selection through cosine similarity.

Specifically, when training task $T_n$, we first train a DT policy $\mu_n$ separately and copy the parameters in the head of $\mu_n$ directly to the corresponding head $\theta_n$. Then, the entire MH-DT is updated by the three-part objective. The first is the action prediction goal of DT.

$$\mathcal{L}_{\text{predict}} = \mathbb{E}_{\tau \sim \mathcal{D}_n} \left( \pi_n(\tau) - a_{target} \right)^2 \tag{1}$$

where $\tau$ is a trajectory sequence from $D_n$ and $a_{target}$ is the last action in $\tau$. Secondly, we use a distillation objective to force $\pi_n$ to be close to $\mu_n$ to enhance learning ability.

$$\mathcal{L}_{\text{distillation}} = \mathbb{E}_{\tau \sim \mathcal{D}_n} \left( \pi_n(\tau) - \mu_n(\tau) \right)^2 + \mathbb{E}_{\tau \sim \mathcal{D}_n} \left( \boldsymbol{H}_{\pi_n} - \boldsymbol{H}_{\mu_n} \right)^2 \tag{2}$$

where $\boldsymbol{H}_{\pi_n}$, $\boldsymbol{H}_{\mu_n}$ denote the hidden states of $\pi_n$ and $\mu_n$ networks respectively, which consist of a sequence of hidden vectors. Such additional distillation loss from intermediate states has been shown to improve results in distilling PLMs (Jiao et al., 2020). The last one is a rehearsal objective, aiming to clone the previous experience in $B_1$ to $B_{n-1}$.

$$\mathcal{L}_{\text{rehearsal}} = \frac{1}{n-1} \sum_{j=1}^{n-1} \mathbb{E}_{\tau \sim \mathcal{B}_j} \left( \pi_j(\tau) - a \right)^2 \tag{3}$$

To address the decreased learning effect caused by multiple review tasks, we propose a selective review mechanism, capitalizing on DT's ability to benefit from training on similar tasks, as discussed in Section 3.2. The task with the lowest similarity is most susceptible to forgetting during the current task's learning process, whereas tasks with greater similarity can benefit from the training on the current task. Specifically, instead of reviewing all the previous tasks $T_1$ to $T_{n-1}$, the similarity between the previous tasks and the current task is measured, and only the K tasks with the smallest

similarity $[T_{s1}, \ldots, T_{s_k}]$ are reviewed. We collect a batch of data from the current training data set $D_n$, calculate the **cosine similarity** of the hidden states of each $\pi_j, j = \{1, \ldots, n-1\}$ and $\mu_n$. These similarities are then sorted, and the indices of the smallest $k$ tasks are compiled into a list, $S$.

$$S = argsort([C_1, \ldots, C_{n-1}])[0:K], \text{ where } C_j = \mathbb{E}_{\tau \sim \mathcal{D}_n} Cosine\_Similarity(\boldsymbol{H}_{\pi_j}, \boldsymbol{H}_{\mu_n}) \quad (4)$$

$K$ is a hyperparameter that determines the number of tasks that need to be reviewed. Then the rehearsal objective in Eq.(3) can be written as:

$$\mathcal{L}_{\text{rehearsal}} = \frac{1}{k} \sum_{j=s_0}^{s_{k-1}} \mathbb{E}_{\tau \sim \mathcal{B}_j} \left( \pi_j(\tau) - a \right)^2 \quad (5)$$

The total loss function for training the whole policy $\pi$ is then,

$$\mathcal{L}_{\text{total}} = \mathcal{L}_{\text{predict}} + \lambda_1 \mathcal{L}_{\text{distillation}} + \lambda_2 \mathcal{L}_{\text{rehearsal}} \quad (6)$$

where $\lambda_1$ and $\lambda_2$ are weights to balance the impact of distilling knowledge about the current task and reviewing previous tasks.

## 4.2 LoRA-DT: Replay Buffer Free CORL

In real-world scenarios, sometimes the replay buffer is not available due to reasons such as data privacy. In order to solve this problem, we propose a method, LoRA-DT, based on the characteristics of DT that does not require a replay buffer to avoid forgetting.

Lawson & Qureshi (2023) investigated the feasibility of directly merging the weights of Decision Transformers (DTs) trained for different tasks to create a multi-task model. Their findings suggest that, for sequential decision-making tasks, DTs rely less on attention and place more emphasis on MLP layers. Low-Rank Adaptation (LoRA) (Hu et al., 2021) adds pairs of rank-decomposition weight matrices (called update matrices) to existing weights and only trains those newly added weights for efficient tuning and avoiding forgetting.

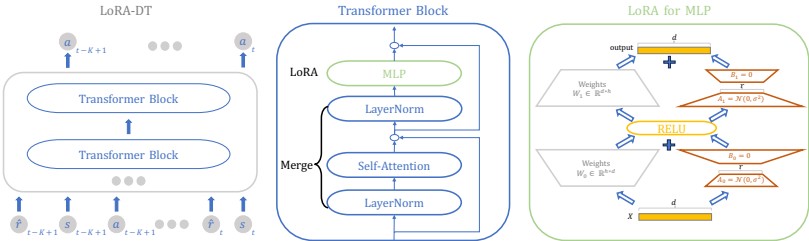

Figure 3: Model architecture of LoRA-DT. In each block of DT, we first fuse and freeze the weights of layers except the MLP layer as in Eq.(7), then use LoRA to fine-tune the MLP layer as in Eq.(9). The rightmost picture is a schematic diagram of LoRA. We fix the original parameter matrix $\boldsymbol{W}_0, \boldsymbol{W}_1$, multiply the two matrices $\boldsymbol{AB}$ to represent the update of the weight matrix, and add it to the original calculation result.

Inspired by the weight property of DT and saved LoRA matrices can be used to avoid catastrophic forgetting, we proposed LoRA-DT to fine-tune the MLP layer in each block of DT through Low-rank adaptation and save the LoRA matrices $\boldsymbol{AB}$ of each task. The schematic diagram of our proposed architecture is given in Fig.3. When rank $r$ is small enough, saving the $\boldsymbol{AB}$ matrices is significantly more memory efficient than saving the replay buffer, and has better ability to prevent catastrophic forgetting. Specifically, when training the first task $T_1$, we train a DT model $\pi$ and update all parameters in it. Then, when training $T_n, n > 1$, we first train a DT model $\mu_n$ separately, and fuse all parameters except the MLP layer in each block with the current model $\pi$ through:

$$\theta_\pi = (1 - \lambda)\theta_\pi + \lambda \theta_{\mu_n} \quad (7)$$

where $\theta$ represents all parameters in DT except the MLP parameters of each block and $\lambda$ is a weight to balance the impact of the merge. Then, we fine-tune MLP layers using LoRA. The form of the original MLP layer is:

$$MLP(\boldsymbol{X}) = \boldsymbol{W}_1(RELU(\boldsymbol{W}_0\boldsymbol{X} + b_0)) + b_1 \quad (8)$$

where $\boldsymbol{W}_0 \in \mathbb{R}^{d \times h}$ and $\boldsymbol{W}_1 \in \mathbb{R}^{h \times d}$ represent the weight of two linear layers, $b$ is bias and $\boldsymbol{X} \in \mathbb{R}^{l \times h}$ is the input of MLP layer. $h, d, l$ respectively represent the hidden dim of the transformer, the inner dim of the MLP layer and the input token length. The form of fine-tuning using LoRA is as follows:

$$MLP(\boldsymbol{X}) = (\boldsymbol{W}_1 + \boldsymbol{B}_1 \boldsymbol{A}_1) RELU((\boldsymbol{W}_0 + \boldsymbol{B}_0 \boldsymbol{A}_0) \boldsymbol{X} + b_0) + b_1 \quad (9)$$

$\boldsymbol{A}_0 \in \mathbb{R}^{r \times h}$, $\boldsymbol{B}_0 \in \mathbb{R}^{d \times r}$, $\boldsymbol{A}_1 \in \mathbb{R}^{r \times d}$ and $\boldsymbol{B} \in \mathbb{R}^{h \times r}$ are update matrices in which $r$ is the rank of LoRA and $r << \min(d, h)$. After training, we save the updated matrices $\mathbb{M}_n = k * [\boldsymbol{A}_0, \boldsymbol{B}_0, \boldsymbol{A}_1, \boldsymbol{B}_1]$ for task $T_n$, where $k$ is the number of blocks in DT. The space occupied by saving update matrices for each task is $2 * k * r * (h + d)$. We only need to change the update matrices to the corresponding $\mathbb{M}_i$ when evaluating task $T_i$.

# 5 EXPERIMENTS

## 5.1 SETUP

**Baselines** In our experiments, we mainly compare the current SOTA method OER and some DT variants. Our baselines are as follows:

- **OER** (Gai et al., 2023): Use a trained model to select experience and a new dual behavior cloning (DBC) architecture to avoid the disturbance of behavior-cloning loss on the Q-learning process. It currently stands as the most proficient method within CORL setting.

- **PDT** (Prompt Decision transformer) (Xu et al.): A multi-task method based on DT. The multi-task learning setting can obtain datasets on all tasks simultaneously so that does not suffer from the catastrophic forgetting problem and can be seen as superior.

- **Vanilla DT** (Cha et al., 2021): We directly apply vanilla DT to the CORL setting for comparison, in order to prove that our method can indeed reduce catastrophic forgetting.

**Offline Sequential Datasets** We consider four sets of tasks from widely-used continuous control environments as in Gai et al. (2023) and Mitchell et al. (2020): Ant-Dir, Walker-Par, Cheetah-Vel and Meta-World reach-v2. For each environment, we randomly sample six tasks to form sequential tasks $T_1$ to $T_6$. To consider different data quality, we selected different time periods in the online training buffer and obtained expert-quality data and middle-quality data as in Mitchell et al. (2020).

**Metrics** Following Lopez-Paz & Ranzato (2017), we adopt the average performance (PER), the backward transfer (BWT) and forward transfer (FWT) as evaluation metrics,

$$\text{PER} = \frac{1}{N} \sum_{n=1}^{N} a_{N,n}, \text{BWT} = \frac{1}{N-1} \sum_{n=1}^{N-1} a_{n,n} - a_{N,n}, \text{FWT} = \frac{1}{N-1} \sum_{n=2}^{N} a_{n-1,n} - \bar{b}_n \quad (10)$$

where $a_{i,j}$ means the final cumulative rewards of task $j$ after learning task $i$ and $\bar{b}_n$ means the test performance for each task at random initialization. For PER, higher is better; for BWT, lower is better; for FWT, higher is better. Lower BWT and higher FWT are preferred when similar PER.

See more details for datasets and metrics in Appendix.B.

## 5.2 OVERALL RESULTS

In this section, we list the performance metrics of all methods under four environments and two data qualities in Table 1. In addition, we also drew the learning curve of each method to more intuitively observe the learning efficiency and forgetting degree in Fig. 4. Due to space limitations, we show the training curve of Walker_Par in the text. See Appendix.C for all experiment results.

Consistently across tasks and data quality, MH-DT outperforms other algorithms for most test experiments in metrics of PER and FWT while LoRA-DT performs outstandingly on the BWT, showing its strong ability to avoid forgetting. Specifically, we can draw the following conclusions.

For the DT structure, all DT-based methods have higher FWT than AC-based method OER, indicating that the DT structure has stronger zero-shot generalization ability when dealing with similar tasks. In some environments such as Ant-Dir, Vanilla DT has a similar PER with OER, while having significantly larger BWT, indicating a better learning effect and more serious catastrophic forgetting.

Table 1: Performance of MH-DT and LoRA-DT compared with baselines

| Dataset Quality | Methods | Ant-Dir | | | Walker-Par | | | Cheetah-Vel | | | Meta-World reach-v2 | | |
|---|---|---|---|---|---|---|---|---|---|---|---|---|---|
| | | PER | BWT | FWT | PER | BWT | FWT | PER | BWT | FWT | PER | BWT | FWT |
| Middle | PDT | 347.9 | - | - | 540.2 | - | - | -39.3 | - | - | - | - | - |
| | OER | 193.6 | 125.6 | -111.8 | 56.0 | 132.9 | 97.3 | -113.8 | 43.8 | 85.1 | - | - | - |
| | Vanilla DT | 146.7 | 248.7 | 102.2 | 380.9 | 166.1 | **331.9** | -118.8 | 99.5 | **154.1** | - | - | - |
| | MH-DT | **326.5** | 38.3 | **110.5** | 510.8 | -32.0 | 314.2 | **-35.4** | **3.4** | 144.6 | - | - | - |
| | LoRA-DT | 230.3 | **5.1** | 98.4 | 424.6 | -2.5 | 321.1 | -57.8 | 3.6 | 150.4 | - | - | - |
| Expert | PDT | 528.8 | - | - | 537.2 | - | - | -17.5 | - | - | 532.3 | - | - |
| | OER | 126.0 | 260.7 | -126.5 | 74.0 | 117.5 | 28.3 | -201.9 | 35.1 | -32.4 | 0.6 | 0.4 | 0.4 |
| | Vanilla DT | 141.7 | 459.7 | 153.8 | 387.4 | 252.2 | **333.1** | -119.8 | 130.9 | 183.8 | 90.0 | 541.1 | **2.3** |
| | MH-DT | **437.8** | 90.8 | **155.2** | 505.4 | 50.4 | 250.8 | **-21.9** | 12.9 | 206.3 | **441.6** | **124.5** | 1.0 |
| | LoRA-DT | 355.1 | **6.4** | 150.9 | 418.6 | **7.1** | 273.2 | -37.3 | **11.5** | **211.3** | 150.2 | 248.9 | 2.0 |

For MH-DT, while keeping the FWT value almost the same as Vanilla DT, MH-DT significantly reduces BWT and thus improves the PER metric, indicating that MH-DT retains the generalization ability of DT and solves catastrophic forgetting by dividing all models into common parts and task-specific parts. In the most difficult Meta-World environment, MH-DT can also learn better-performing strategies, while OER cannot learn any usable strategies at all. It is also exciting to find that MH-DT can achieve comparable PER to the upper bound PDT.

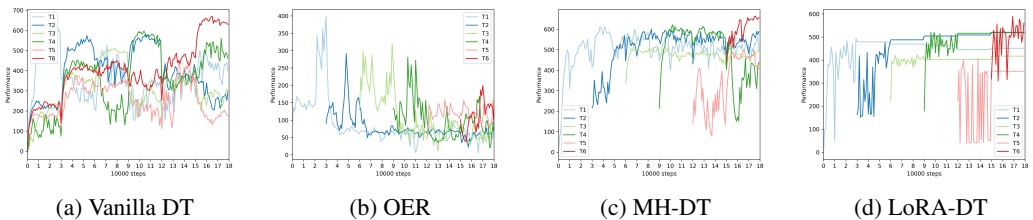

| (a) Vanilla DT | (b) OER | (c) MH-DT | (d) LoRA-DT |

Figure 4: Process of learning six sequential tasks in Walker_Param, where our methods MH-DT and LoRA-DT are compared with two baselines Vanilla DT and OER. Every 30K steps on one task.

For LoRA-DT, as in Fig. 4d, the performance of LoRA-DT in previous tasks after the training phase is completely unchanged except for a small fluctuation in the merge phase at the beginning of each new task. However, even though LoRA-DT demonstrates the highest BWT level, indicating its exceptional ability to prevent forgetting, its performance in PER does not surpass that of MH-DT. This can be attributed to the fact that the fine-tuning method, merge+LoRA, is still not on par with the approach of updating all parameters. It's also worth noting that in an environment where the FWT is larger, signifying greater task similarity, the performance of LoRA-DT is not much behind that of MH-DT as in Walker_Par and Cheetah_Vel, but relatively larger in other environments.

## 5.3 ABLATION STUDY

### 5.3.1 DISTILLATION OBJECTIVE AND SELECTIVELY REHEARSAL IN MH-DT

In MH-DT, we propose an additional distillation objective and selective rehearsal to make the current policy $\pi_n$ closer to our learned teacher policy $\mu_n$, which also means getting better performance on the current task. In order to verify the effectiveness of the two modules, distillation objective (DO) and selective rehearsal (SR), we do ablation studies by comparing MH-DT with OER and three variants: MH-DT without distillation objective (MH-DT w/o DO), MH-DT without selective rehearsal (MH-DT w/o DO), MH-DT without distillation objective and selective rehearsal (MH-DT w/o DO w/o SR) on distillation gap (DG), a new metric to measure the gap between the task-specific policy $\mu_n$ that is trained separately and the current policy $\pi_n$ after training process on task $T_n$.

$$\text{DG} = \frac{1}{N} \sum_{n=1}^{N} t_n - a_{n,n} \tag{11}$$

where $t_n$ is the final cumulative reward of $\mu_n$, and $a_{n,n}$ is the performance of task $n$ after learning task $n$. A smaller DG indicates a smaller gap and a stronger ability to learn the current task.

Table 2: Ablation on distillation objective (DO) and selective rehearsal (SR) in MH-DT

| Methods | Ant-Dir | | Walker-Par | | Cheetah-Vel | |
|---|---|---|---|---|---|---|
| | PER | DG | PER | DG | PER | DG |
| OER | 193.6 | 53.3 | 56.0 | 417.9 | -113.8 | 26.5 |
| MMH-DT w/o DO w/o SR | 223.9 | 41.1 | 374.8 | 231.7 | -86.1 | 9.8 |
| MH-DT w/o DO | 310.3 | 30.3 | 456.2 | 90.5 | -54.7 | -4.1 |
| MH-DT w/o SR | 302.5 | 37.3 | 423.0 | 125.2 | -64.1 | 0.3 |
| MH-DT | 326.5 | -4.0 | 510.8 | 54.1 | -35.4 | -21.6 |

From Table 2, we can observe that: 1)DO and SR can both improve MH-DT's ability to learn current tasks. 2) The DT-based method works better than the AC-based method When using action cloning to transfer knowledge. 3)By correctly selecting reviewed tasks, the performance of the current task can even exceed that of the teacher policy $\mu_n$ as in Cheetah_Vel.

### 5.3.2 RANK r AND BUFFER SIZE

We compare the space occupied performance and metrics performance of OER, MH-DT and LoRA-DT under different rank $r$ and different sizes of replay buffer. In this section, we change the buffer size to 1K, 3K, and 10K and rank $r$ to 4, 16, 64. In order to intuitively compare the size of the occupied space, we take the space of 10K sample buffer as the benchmark and record it as 100%. It is worth noting that because different environments have different state-action dimensions, the proportion of memory occupied by LoRA-DT is also different.

Table 3: Performance when buffer size and rank change

| Methods | Ant-Dir | | | Walker-Par | | | Cheetah-Vel | | | Meta-World reach-v2 | | |
|---|---|---|---|---|---|---|---|---|---|---|---|---|
| | PER | BWT | Memory | PER | BWT | Memory | PER | BWT | Memory | PER | BWT | Memory |
| OER(1K) | 126.0 | 260.7 | 10% | 74.0 | 117.5 | 10% | -201.9 | 35.1 | 10% | 0.6 | 0.4 | 10% |
| OER(3K) | 150.2 | 234.2 | 30% | 80.5 | 114.3 | 30% | -184.2 | 31.1 | 30% | 0.6 | 0.4 | 30% |
| OER(10K) | 187.3 | 198.5 | 100% | 96.2 | 110.5 | 100% | -169.1 | 24.2 | 100% | 0.6 | 0.4 | 100% |
| MH-DT(1K) | 437.8 | 90.8 | 10% | 505.4 | 50.4 | 10% | -21.9 | 12.9 | 10% | 441.6 | 124.5 | 10% |
| MH-DT(3K) | 454.1 | 74.3 | 30% | 515.7 | 32.1 | 30% | -18.4 | 6.3 | 30% | 463.8 | 100.2 | 30% |
| MH-DT(10K) | 490.6 | 43.2 | 100% | 537.2 | 12.5 | 100% | -16.3 | 2.2 | 100% | 500.3 | 88.1 | 100% |
| LoRA-DT(r=4) | 355.1 | 6.4 | 1.6% | 418.6 | 7.1 | 2.4% | -37.3 | 11.5 | 2.2% | 150.2 | 148.9 | 1.3% |
| LoRA-DT(r=16) | 387.4 | 6.7 | 6.6% | 427.3 | 6.9 | 9.8% | -30.5 | 10.3 | 8.7% | 162.8 | 132.2 | 5.4% |
| LoRA-DT(r=64) | 407.54 | 5.8 | 26.5% | 469.2 | 7.1 | 39.3% | -33.4 | 11.3 | 35.1% | 168.5 | 149.5 | 21.8% |

The results in Table 3 demonstrate the memory efficiency of LoRA-DT, it can achieve performance that exceeds OER and is close to MH-DT while using nearly one-tenth of the space used by them. In addition, for methods that are based on ER, a larger buffer size can significantly reduce the BWT value, thereby improving performance. When the size of the replay buffer approaches the training dataset, it will become a multi-task problem. On the contrary, LoRA-DT's forgetting metric BWT is not sensitive to the rank $r$. However increasing $r$ can increase the plasticity of the LoRA-DT model, allowing it to obtain better-performing policies through fine-tuning. When $r$ increases to the inner dim of the MLP, the tuning effect is equivalent to directly fine-tuning the MLP layer.

## 6 CONCLUSION

In this work, we rethink the CORL problem through Decision Transformer (DT), highlighting the advantages of DT over AC-structured algorithms and focusing on addressing the more severe forgetting issue. Subsequently, we introduce MH-DT which employs multiple heads to store task-specific knowledge, facilitates knowledge sharing with a common component, and incorporates distillation and selective rehearsal modules to enhance learning capacity. When replay buffers are unavailable, we propose LoRA-DT, which merges impactful weights for knowledge sharing and fine-tunes the crucial MLP layer within DT blocks using LoRA. Experiments and analysis show that our DT-based methods outperform SOTA baselines on various continuous control tasks.

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

# A ADDITIONAL ALGORITHM DESCRIPTIONS

## A.1 PSEUDO-CODE

We present the algorithms for MH-DT training in Sec.A.1.1 and for LoRA-DT in Sec.A.1.2

### A.1.1 MH-DT

---
**Algorithm 1** MH-DT training
---
**Input**: Number of task N; Number of select task K; Number of select frequency T; Dataset $D_i$ of each task $T_i$, $i \in [1, \ldots, N]$; Initial the policy $\pi$.

1: **for** Tasks $T_n$ in $[1, \ldots, N]$ **do**
2:     Get dataset $D_n$; Get replay buffers for previous task $B_1, \ldots, B_{n-1}$; Initial the replay buffer $B_n = \emptyset$; Initial new head $h_n$ for $\pi$; Initial DT policy $\mu_n$; Initial select set $S = [1, \ldots, N-1]$.

3:     **for** step $i$ in $range(max\_steps)$ **do**
4:         Update $\mu_n$ via minimizing mean-squared loss.
5:         Copy the parameters in head of $\mu_n$ to head $h_n$.
6:         **if** $i \mod T == 0$ **then**
7:             Select $K$ tasks and add them to $S$ via Eq.(4).
8:         **end if**
9:         Sample a batch from $D_n$.
10:        Calculate $\mathcal{L}_{\text{predict}}$ with mean-squared loss.
11:        Calculate $\mathcal{L}_{\text{distillation}}$ via Eq.(2).
12:        **for** $j$ in $S$ **do**
13:            Sample a batch from $B_j$.
14:            Calculate $\mathcal{L}_{\text{rehearsal}}$ with corresponding $\pi_j$ as in Eq.(5).
15:        **end for**
16:        Update $\pi$ with $\mathcal{L}_{\text{total}}$ in Eq.(6).
17:     **end for**
18:     Randomly select trajectories in $D_n$ and add them to $B_n$.
19: **end for**
**Output**: Policy $\pi$.

---

### A.1.2 LoRA-DT

---
**Algorithm 2** LoRA-DT training
---
**Input**: Number of task N; Number of blocks k; Dataset $D_i$ of each task $T_i$, $i \in [1, \ldots, N]$; Initial the policy $\pi$.

1: **for** Tasks $T_n$ in $[1, \ldots, N]$ **do**
2:     Get dataset $D_n$; Get replay buffers for previous task $B_1, \ldots, B_{n-1}$; Initial the replay buffer $B_n = \emptyset$; Initial DT policy $\mu_n$; Initial update matrices $\boldsymbol{AB}$; Initial $\mathbb{M}_n = \emptyset$.
3:     **if** n == 1 **then**
4:         Update $\pi$ via minimizing mean-squared loss.
5:     **else**
6:         Update $\mu_n$ via minimizing mean-squared loss.
7:         Merge the parameters in $\mu_n$ to $\pi$ except for the MLP layers of each block as in Eq.(7).
8:         Update matrices $\boldsymbol{AB}$ with LoRA as in Eq.(9)
9:     **end if**
10:    Randomly select trajectories in $D_n$ and add them to $B_n$.
11:    Save $\mathbb{M}_n = k * [\boldsymbol{A}_0, \boldsymbol{B}_0, \boldsymbol{A}_1, \boldsymbol{B}_1]$ for $T_n$
12: **end for**
**Output**: Policy $\pi$.

---

## A.2 HYPERPARAMETERS

We show the common hyperparameters of DT in Table.4 and specific hyperparameters of our MH-DT and LoRA-DT in Table.5 and Table.6.

Table 4: Common Hyperparameters of all Decision Transformer

| Hyperparameters | Value |
|---|---|
| $K$ (length of context $\tau$ ) | 20 |
| number of evaluation episodes for each task | 10 |
| learning rate | $1e-4$ |
| learning rate decay weight | $1e-4$ |
| number of layers | 3 |
| number of attention heads | 1 |
| embedding dimension | 128 |
| activation | ReLU |

Table 5: Specific Hyperparameters of MH-DT

| Hyperparameters | Value |
|---|---|
| number of select task K | 2 |
| select frequency T | 10 |
| replay buffer size | 1K |
| weight of distillation $\lambda_1$ | 0.5 |
| weight of distillation $\lambda_2$ | 1.0 |

Table 6: Specific Hyperparameters of LoRA-DT

| Hyperparameters | Value |
|---|---|
| rank $r$ | 4 |
| inner dim of the MLP layer | 128 |
| weight of merge $\lambda$ | 0.2 |

## B EXPERIMENT DETAILS

We present the detail of offline sequential datasets in Sec.B.1, details of metrics in Sec.B.2 and implement details in Sec.B.3.

### B.1 OFFLINE SEQUENTIAL DATASETS

We consider four sets of tasks from widely-used continuous control environments as in Gai et al. (2023) and Mitchell et al. (2020):

- Ant-2D Direction (Ant-Dir): train a simulated ant with 8 articulated joints to run in a 2D direction

- Walker-2D Params (Walker-Par): train a simulated agent to move forward, where different tasks have different parameters. Specifically, different tasks require the agent to move at different speeds

- Half-Cheetah Velocity (Cheetah-Vel): train a cheetah to run at a random velocity. Cheetah-vel is unique in that as the 'vel' number increases the task becomes more challenging.

- Meta-World reach-v2. Tasks are to control a Sawyer robot's end-effector to reach different target positions in 3D space. The agent directly controls the XYZ location of the end-effector.

For Ant-Dir, Walker-Par and Meta-World reach-v2, we randomly sample six tasks to form sequential tasks $T_1$ to $T_6$. For Cheetah-Vel, we fixedly select the six tasks of vel=3, 6, 9, 12, 15, 18 and train them in order. The difficulty of the six tasks increases in sequence.

To consider different data quality, we selected different time periods in the online training buffer and obtained expert-quality data and middle-quality data as in Mitchell et al. (2020). For Meta-World reach-v2, we use the expert dataset because only expert script policies are available.

## B.2 METRICS

use evaluation metrics to evaluate the continuous learning ability of the algorithm as in Lopez-Paz & Ranzato (2017). More specifically, we would like to measure:

• *Average Performance* (PER), which measures average performance on all tasks after training.

$$\text{PER} = \frac{1}{N} \sum_{n=1}^{N} a_{N,n} \tag{12}$$

• *Backward transfer* (BWT), which is the influence that learning a task $t$ has on the performance on a previous task $k \prec t$. On the one hand, there exists positive backward transfer when learning about some task $t$ increases the performance on some preceding task $k$. On the other hand, there exists negative backward transfer when learning about some task $t$ decreases the performance on some preceding task $k$. Large negative backward transfer is also known as catastrophic forgetting.

$$\text{BWT} = \frac{1}{N-1} \sum_{n=1}^{N-1} a_{n,n} - a_{N,n} \tag{13}$$

• *Forward transfer* (FWT), which is the influence that learning a task $t$ has on the performance on a future task $k \succ t$. In particular, positive forward transfer is possible when the model is able to perform "zero-shot" learning, perhaps by exploiting the structure available in the task descriptors.

$$\text{FWT} = \frac{1}{N-1} \sum_{n=2}^{N} a_{n-1,n} - \bar{b}_n \tag{14}$$

where $a_{i,j}$ means the final cumulative rewards of task $j$ after learning task $i$ and $\bar{b}_n$ means the test performance for each task at random initialization. For PER, higher is better; for BWT, lower is better; for FWT, higher is better. If two models have similar PER, the most preferable one is the one with lower BWT and higher FWT.

## B.3 IMPLEMENT DETIALS

For each evaluation step, we test all strategies on the corresponding tasks 10 times and report the average.

For LoRA-DT, the MLP layer in each block in the original DT implementation is composed of two Cov1D layers plus an intermediate activation layer. In our implementation, we replace the Cov1D layer with a liner class [1] which contains a LoRA Layer and a Linear layer. Update the parameters of the Linear layer when training the first task $T_1$, and then only fine-tune the LoRA matrix $\boldsymbol{AB}$ and save it after training each task.

## C   ADDITIONAL RESULTS

We present the training curves for all environments and dataset qualities in this section.

## C.1   TRAING CURVES

---

[1] https://github.com/microsoft/LoRA

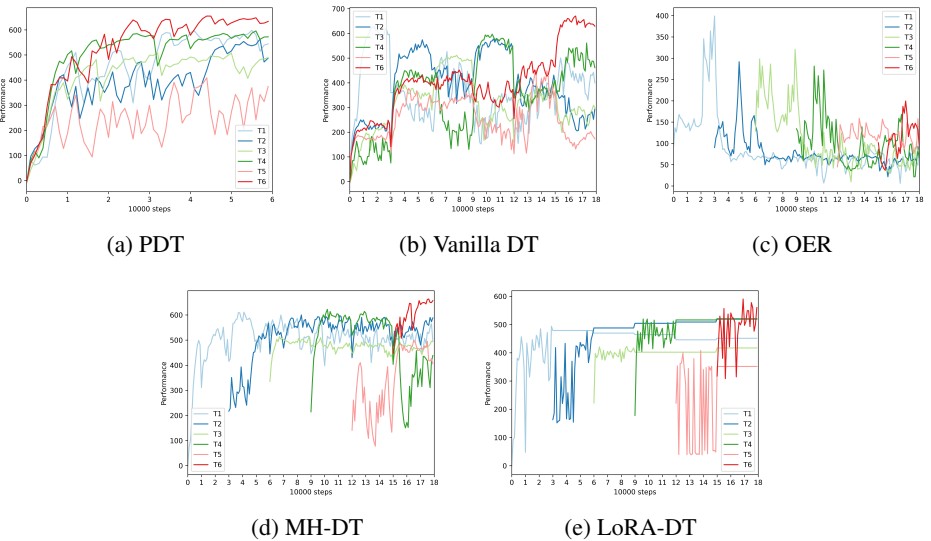

(a) PDT  (b) Vanilla DT  (c) OER

(d) MH-DT  (e) LoRA-DT

Figure 5: Walker_Param (middle)

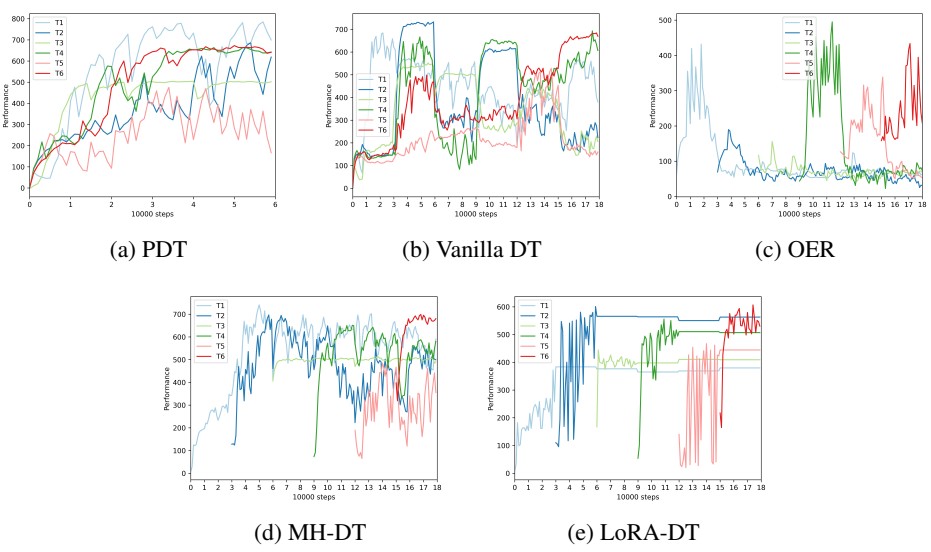

(a) PDT  (b) Vanilla DT  (c) OER

(d) MH-DT  (e) LoRA-DT

Figure 6: Walker_Param (expert)

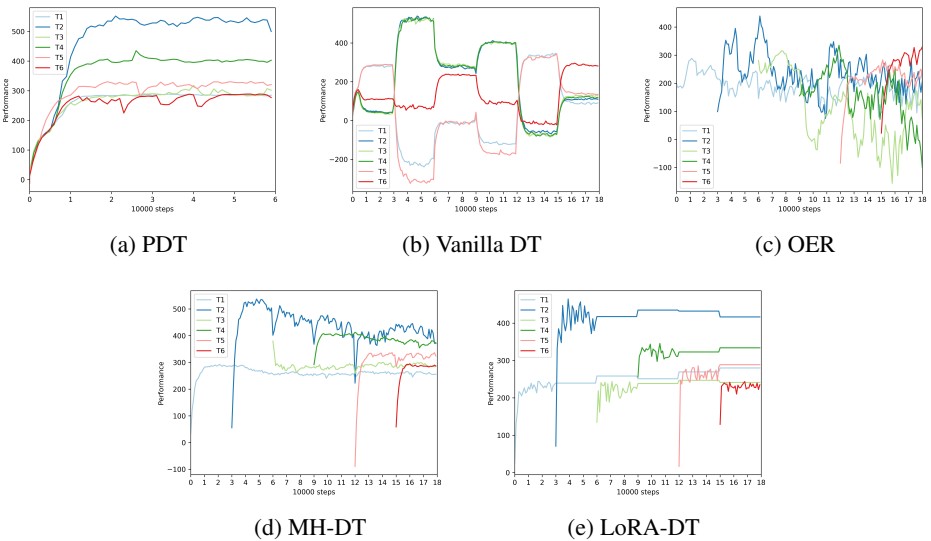

Figure 7: Ant_Dir (middle)

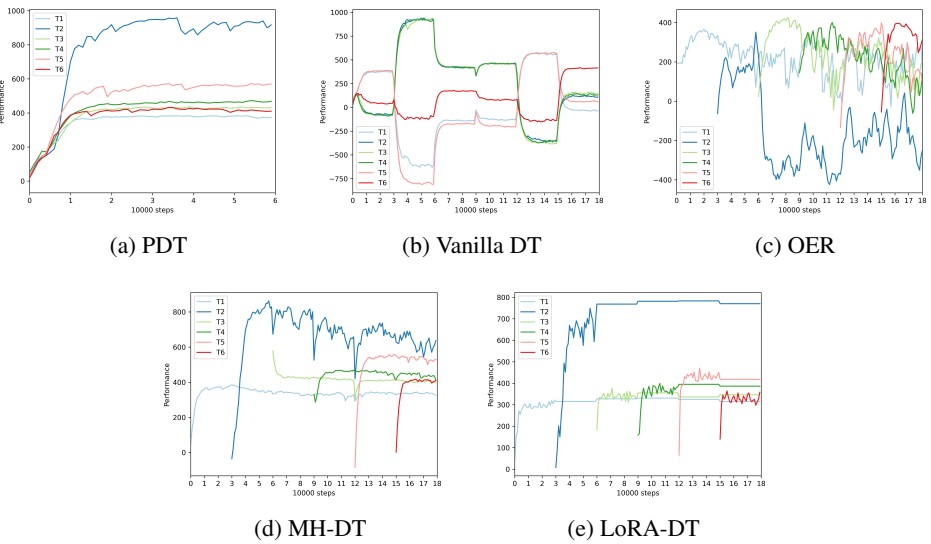

Figure 8: Ant_Dir (expert)

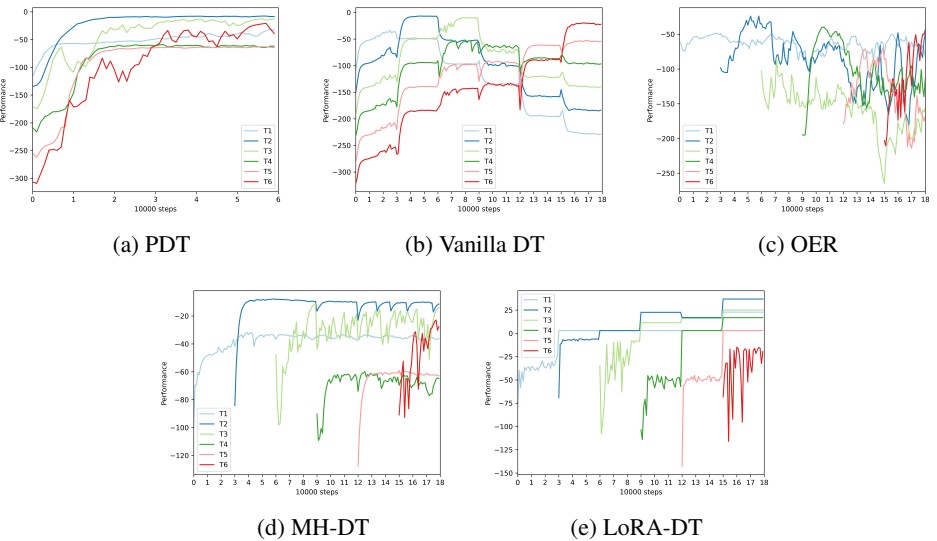

(a) PDT      (b) Vanilla DT      (c) OER

(d) MH-DT      (e) LoRA-DT

Figure 9: Cheetah_Vel (middle)

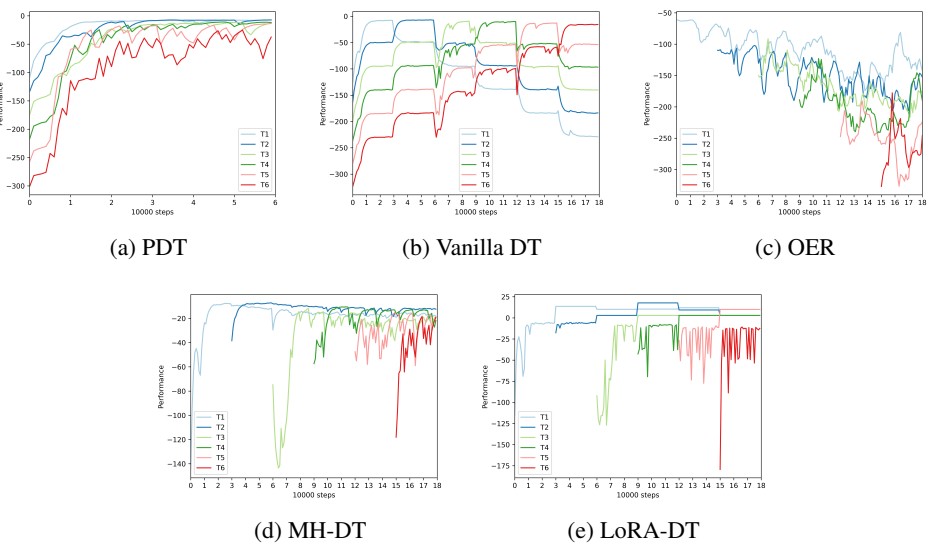

(a) PDT      (b) Vanilla DT      (c) OER

(d) MH-DT      (e) LoRA-DT

Figure 10: Cheetah_Vel (expert)

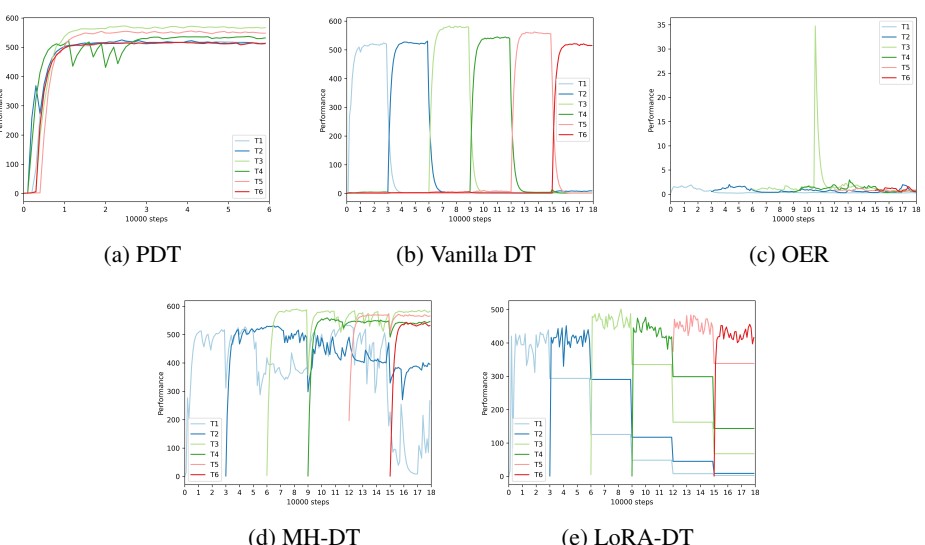

(a) PDT  (b) Vanilla DT  (c) OER

(d) MH-DT  (e) LoRA-DT

Figure 11: ML1-pick-place-v2

