# OpenReview forum: "Solving Continual Offline Reinforcement Learning with Decision Transformer"
_ICLR.cc/2024/Conference — ICLR 2024 Conference Withdrawn Submission_

### Official Review · Reviewer_Md6w · 2023-10-24

**Soundness:** 2 fair
**Presentation:** 1 poor
**Contribution:** 2 fair
**Rating:** 3
**Confidence:** 4

**Summary:**

This paper proposes two methods - MH-DT and LoRA-DT - to mitigate catastrophic forgetting in continuous offline reinforcement learning using Decision Transformers. MH-DT uses multiple heads and selective replay while LoRA-DT merges weights and adapts with low-rank matrices. Experiments on MuJoCo and Meta-World benchmarks demonstrate performance gains over prior methods.

**Strengths:**

- Utilizing Decision Transformers is an interesting direction for offline continual RL.
- The idea of using LoRA for efficient adaptation while avoiding forgetting is logical and well-motivated.

**Weaknesses:**

1. The experimental evaluation is very limited, with results reported on just a few MuJoCo and Meta-World environments. Important details like the number of random seeds used, variance across seeds, and standard errors are missing. With such sparse experiment details and a lack of reported variance, it is difficult to assess the statistical significance of the results and conclusions. More environments, hyperparameter configurations, and random seeds should be evaluated to substantiate the claims.
2. The comparison to other continual learning techniques is lacking. The paper only considers an offline RL baseline but does not compare against many strong regularization-based continual learning methods like [1,2,3]. Additionally, more offline RL algorithms could be included as baselines beyond just DT variants. The restrictive set of baselines makes it hard to situate the performance of the proposed techniques.
3. Many important implementation details are unclear, like how the datasets are sequentially provided, how rewards are evaluated, the training steps, etc. The hyperparameter values seem arbitrary without proper justification or ablation studies. For instance, the number of training steps likely affects final performance significantly but the impact is not analyzed.

Therefore, the experimental analysis is currently too limited and sparse to demonstrate convincing improvements over existing approaches. A much more thorough evaluation is needed across more environments, seeds, baselines, and design choices to substantiate the conclusions.

[1] Kirkpatrick, James, et al. "Overcoming catastrophic forgetting in neural networks." Proceedings of the national academy of sciences 114.13 (2017): 3521-3526.

[2] Lyle, Clare, Mark Rowland, and Will Dabney. "Understanding and preventing capacity loss in reinforcement learning." arXiv preprint arXiv:2204.09560 (2022).

[3] Xu, Mengdi, et al. "Hyper-decision transformer for efficient online policy adaptation." arXiv preprint arXiv:2304.08487 (2023).

**Questions:**

Some questions remain about the techniques:

1. It is unclear why LoRA and multi-heads differ in their backward transfer capability if they both avoid updating the base model weights. More analysis is needed.
2. The methods of LoRA and multi-heads seem to rely on the base model is well pretrained, such that the pretrained features are rich enough for downstream adaptation, but the paper does not describe any pretraining. This makes the effectiveness unclear. More generally, an ablation study on the base model initialization could provide insights into how reliant the techniques are on pretraining vs. continual adaptation.

---

### Official Review · Reviewer_ZQbc · 2023-10-31

**Soundness:** 2 fair
**Presentation:** 2 fair
**Contribution:** 2 fair
**Rating:** 3
**Confidence:** 4

**Summary:**

The paper proposed an algorithm for the continuous offline reinforcement learning to address the issues like distribution shifts, low efficiency, and weak knowledge-sharing in the Actor-Critic structures. They used multi-head decision-transformers and LoRA adapter to mitigate forgetting problem.

**Strengths:**

- The method has application to the real-world problems.

**Weaknesses:**

- The paper is not well-written. It is not clear what issue they want to address specifically. I do do not find the story of the paper well connected across the different parts. Specially what they said in the abstract is not aligned exactly with what they claim in the experiment part.
- the paper does not have enough innovation.
- The related work does not have actually the recent related works. There are many works with DT for decision environment which are not in the related works.
- Also I should mention the baseline are not fair! First in the abstract they compared their work with actor-critic while in the experiments they did not consider this as a baseline to compare with. In addition to that they need to compare their work with recent works using DT in offline RL.
- The plots are not shown in a good way to present the conclusion.
- More ever, the results in the tables are not consistent with the claims in the paper and the experiment section.

**Questions:**

- What is exactly the proposed algorithm wants to solve which other works did not?
- What does selective rehearsal do exactly?
- What do you mean by zero-shot generalization in section 3 and how did you do that with supervised training?

---

### Official Review · Reviewer_b7vq · 2023-11-01

**Soundness:** 3 good
**Presentation:** 3 good
**Contribution:** 2 fair
**Rating:** 5
**Confidence:** 2

**Summary:**

The author proposes to extend DT to the problem of continuous learning. In order to adapt the DT, the author propose MHDT for replay buffer version and Lora DT without replay buffer. Comparing with baselines, the proposed methods prevents catastrophic forgetting.

**Strengths:**

The paper propose an idea of adapting DT in the context of continuous learning, which is novel. The proposed methods work compare with vanilla DT and previous baseline such as OER. The results seems solid.

**Weaknesses:**

While I am not an expert in continuous RL evaluation, I do think the paper needs to show the results for different task ordering, which would make the results more convincing.

The other major concern for me is I don't know how relevant OER is as a baseline, it seems to be pretty old.

What about traditional contunual learning baselines like EWC? Why are they not discussed here since by applying DT, it seems we are getting closer to the supervised settings where a bunch of tricks can be tested.

**Questions:**

See above

---

### Official Review · Reviewer_1HUt · 2023-11-04

**Soundness:** 3 good
**Presentation:** 2 fair
**Contribution:** 2 fair
**Rating:** 5
**Confidence:** 3

**Summary:**

This paper integrates Decision Transformer into the continuous offline reinforcement learning, and also introduce multi-head DT and low-rank adaptation to mitigate DT's forgetting problem. Experimental results show improved learning capabilities and better memory efficiency.

**Strengths:**

1) It's novel to integrate Decision Transformer within the CORL framework.

2) It's novel and interesting to introduce multi-head DT and low-rank adaptation DT to mitigate DT's forgetting problem.

3) Experiments and ablation studies are extensive.

**Weaknesses:**

1) Decision Transformer and Trajectory Transformer (TT) are two very similar work, the paper doesn't show the comparison of DT and TT are integrated into the CORL framework.

2) The performance comparison in Table 1is not very clear.

**Questions:**

Could Trajectory transformer also be integrated into the CORL framework? If it is, how is the performance comparison with DT within CORL?